# Modeling Epithelial Homeostasis and Perturbation in Three-Dimensional Human Esophageal Organoids

**DOI:** 10.3390/biom14091126

**Published:** 2024-09-05

**Authors:** Masataka Shimonosono, Masaki Morimoto, Wataru Hirose, Yasuto Tomita, Norihiro Matsuura, Samuel Flashner, Mesra S. Ebadi, Emilea H. Okayasu, Christian Y. Lee, William R. Britton, Cecilia Martin, Beverly R. Wuertz, Anuraag S. Parikh, Uma M. Sachdeva, Frank G. Ondrey, Venkatram R. Atigadda, Craig A. Elmets, Julian A. Abrams, Amanda B. Muir, Andres J. Klein-Szanto, Kenneth I. Weinberg, Fatemeh Momen-Heravi, Hiroshi Nakagawa

**Affiliations:** 1Herbert Irving Comprehensive Cancer Research Center, Columbia University Irving Medical Center, New York, NY 10032, USA; k5541938@kadai.jp (M.S.); mm6093@cumc.columbia.edu (M.M.); wh2564@cumc.columbia.edu (W.H.); yt2795@cumc.columbia.edu (Y.T.); mm5075mn@yahoo.co.jp (N.M.); sf3070@cumc.columbia.edu (S.F.); mse2144@barnard.edu (M.S.E.); eho2122@barnard.edu (E.H.O.); cl4316@columbia.edu (C.Y.L.); wrb2120@cumc.columbia.edu (W.R.B.); cm4194@cumc.columbia.edu (C.M.); asp2145@cumc.columbia.edu (A.S.P.); ja660@cumc.columbia.edu (J.A.A.); fm2540@cumc.columbia.edu (F.M.-H.); 2Organoid & Cell Culture Core, Columbia University Digestive and Liver Diseases Research Center, New York, NY 10032, USA; 3Department of Otolaryngology, Head and Neck Surgery, Masonic Cancer Center, University of Minnesota, Minneapolis, MN 55455, USA; knier003@umn.edu (B.R.W.); ondre002@umn.edu (F.G.O.); 4Department of Otolaryngology, Head and Neck Surgery, Columbia University, New York, NY 10032, USA; 5Division of Thoracic Surgery, Massachusetts General Hospital, Boston, MA 02114, USA; uma.sachdeva@mgh.harvard.edu; 6Department of Dermatology, University of Alabama, Birmingham, AL 35294, USA; venkatra@uab.edu (V.R.A.); celmets@uabmc.edu (C.A.E.); 7Division of Digestive and Liver Diseases, Department of Medicine, Columbia University Irving Medical Center, New York, NY 10032, USA; 8Division of Gastroenterology, Hepatology and Nutrition, Department of Pediatrics, Children’s Hospital of Philadelphia, University of Pennsylvania Perelman School of Medicine, Philadelphia, PA 19104, USA; muira@chop.edu; 9Histopathology Facility, Fox Chase Cancer Center, Philadelphia, PA 19111, USA; andres.klein-szanto@fccc.edu; 10Department of Pediatrics, Maternal & Child Health Research Institute, Stanford Cancer Institute, Stanford University, Stanford, CA 94305, USA; kw1@stanford.edu; 11Cancer Biology and Immunology Laboratory, College of Dental Medicine, Columbia University Irving Medical Center, New York, NY 10032, USA

**Keywords:** esophagus, organoids, epidermal growth factor, transforming growth factor-β, basal cell hyperplasia, eosinophilic esophagitis, retinoic acid

## Abstract

**Background:** Esophageal organoids from a variety of pathologies including cancer are grown in Advanced Dulbecco’s Modified Eagle Medium-Nutrient Mixture F12 (hereafter ADF). However, the currently available ADF-based formulations are suboptimal for normal human esophageal organoids, limiting the ability to compare normal esophageal organoids with those representing a given disease state. **Methods:** We have utilized immortalized normal human esophageal epithelial cell (keratinocyte) lines EPC1 and EPC2 and endoscopic normal esophageal biopsies to generate three-dimensional (3D) organoids. To optimize the ADF-based medium, we evaluated the requirement of exogenous epidermal growth factor (EGF) and inhibition of transforming growth factor-(TGF)-β receptor-mediated signaling, both key regulators of the proliferation of human esophageal keratinocytes. We have modeled human esophageal epithelial pathology by stimulating esophageal 3D organoids with interleukin (IL)-13, an inflammatory cytokine, or UAB30, a novel pharmacological activator of retinoic acid signaling. **Results:** The formation of normal human esophageal 3D organoids was limited by excessive EGF and intrinsic TGFβ-receptor-mediated signaling. Optimized HOME0 improved normal human esophageal organoid formation. In the HOME0-grown organoids, IL-13 and UAB30 induced epithelial changes reminiscent of basal cell hyperplasia, a common histopathologic feature in broad esophageal disease conditions including eosinophilic esophagitis. **Conclusions:** HOME0 allows modeling of the homeostatic differentiation gradient and perturbation of the human esophageal epithelium while permitting a comparison of organoids from mice and other organs grown in ADF-based media.

## 1. Introduction

Stratified squamous epithelium of the esophagus comprises the basal and the suprabasal compartments. The basal compartment contains proliferative basal/parabasal epithelial cells (keratinocytes) that undergo post-mitotic terminal differentiation within the suprabasal cell layers. Differentiating keratinocytes establish cell junctions to provide epithelial barrier function while undergoing desquamation into the lumen at the superficial cell layers, allowing continuous esophageal epithelial renewal [1,2].

The homeostatic proliferation–differentiation gradient of the esophageal epithelium is subjected to regulation by various factors and conditions including growth factors, hormones, and cell–cell contact. Among these factors are epidermal growth factor (EGF), transforming growth factor (TGF)-β, bone morphogenetic proteins (BMP), Notch, and retinoids/vitamins (see Table 1 and Table 2 for abbreviations). EGF and other ligands for epidermal growth factor receptor (EGFR) stimulate keratinocyte proliferation [3,4,5,6,7,8,9,10]. All-trans retinoic acid, a key metabolite of vitamin A, stimulates basal cell proliferation to increase progenitor cells committed to terminal differentiation [11,12,13]. During terminal differentiation, Notch signaling activates transcription of differentiation-related genes such as Involucrin (IVL) [3,14,15]. Belonging to the TGFβ super family, BMP promotes squamous cell differentiation through activation of the antioxidant response pathway [16]. These pathways are perturbed in a variety of esophageal disease conditions such as eosinophilic esophagitis (EoE), whose pathogenesis involves disrupted mucosal integrity and barrier functions as manifested by basal cell hyperplasia (BCH), a histopathologic feature characterized by expansion of basal cells and delayed terminal differentiation [17,18,19].

The 3D organoid system is a robust cell culture system where a small number of tissue-dissociated cells (1000–10,000) are embedded in the Matrigel^®^ matrix, the basement-membrane extract, to form single cell-derived spherical structures that arise within 2 weeks [20,21]. 3D organoids are amenable to genetic and pharmacological modifications and are utilized to explore how pathogenic agents such as alcohol and pro-inflammatory cytokines may influence esophageal epithelial homeostasis [22]. Patient-derived organoids (PDO) are translatable in personalized medicine to predict therapeutic response and explore novel therapeutics, albeit limited to oncology to date [23,24,25,26,27,28,29].

To date, ADF is the most widely utilized base medium to generate organoids representing various types of normal and abnormal human epithelial conditions, including esophageal intestinal metaplasia, preneoplasia, and cancers [21,23,30]. We have previously described that ADF is suboptimal to grow normal human esophageal organoids and proposed an alternative KSFMC medium with a final concentration of 0.6 mM Ca2+ added to keratinocyte serum-free medium (KSFM) that allows recapitulation of not only the normal squamous epithelial differentiation gradient but also pathologic changes such as BCH [22]. Largely distinct from ADF-based formations, however, KSFMC limits a direct comparison of esophageal organoid culture products such as those representing neoplastic human esophageal conditions and murine counterparts grown in ADF-based media.

Herein, we have optimized the ADF-based medium to grow normal human esophageal organoids, finding that excessive EGF and intrinsic TGFβ-receptor-mediated signaling may be detrimental for normal human esophageal epithelial formation in 3D organoids. In HOME0, our optimized ADF-based medium, biopsy-derived normal human esophageal organoids display morphology and growth kinetics that are comparable to those grown in KSFMC and display BCH-like changes in response to interleukin (IL)-13, an inflammatory cytokine essential in the pathogenesis of eosinophilic esophagitis (EoE) and UAB30, a novel pharmacological activator of retinoic acid signaling.

## 2. Materials and Methods

### 2.1. Cell Culture and 3D Organoid Culture

All the cell culture reagents were purchased from Thermo Fisher Scientific (Waltham, MA, USA) unless otherwise noted. The live cells were determined by Countess^™^ Automated Cell Counter following exclusion of Trypan Blue dye-stained dead cells. Immortalized normal human esophageal cell lines EPC1 and EPC2 were grown in a monolayer culture in Keratinocyte SFM (KSFM) and utilized to generate 3D organoids as described previously [22,31,32]. For both EPC1 and EPC2, we have validated the mycoplasma negative status by the Universal Mycoplasma Detection kit (ATCC, Manassas, VA, USA), and the cell authentication by short tandem repeat profiling (ATCC).

Three independent patient-derived organoid (PDO) lines EN1–3 were established from esophageal normal mucosa biopsies that were obtained via upper endoscopy from an adult patient (EN1) at Columbia University Irving Medical Center (CUIMC) and pediatric patients (EN2 and EN3) at the Children’s Hospital of Philadelphia (CHOP). All the clinical materials were procured from informed-consented patients according to the Institutional Review Board standard and the guidelines of each respective institution. The biopsies at CUIMC were immediately subjected to tissue dissociation to initiate organoid formation on the day of the tissue procurement. The biopsies from patients EN2 and EN3 were frozen in a cryogenic vial containing 90% fetal calf serum (FBS) and 10% Dimethyl sulfoxide (DMSO) (Sigma-Aldrich, St. Louis, MO, USA) and shipped on dry ice to the Organoid and Cell Culture Core (OCCC) facility at the CUIMC and stored in liquid nitrogen until use.

To generate the 3D organoids, an esophageal biopsy was dissociated into a single-cell suspension as described [32]. The tissue-derived cells or cultured cells (EPC1 and EPC2) were seeded into 24-well plates in 50 µL Matrigel at 5000 cells per well, and grown for 11 days with 500 µL (per well) of advanced DMEM/F12 (ADF)-based medium containing 1X GlutaMAX, 1X HEPES, 1X N2 Supplement, 1X B27 Supplement, 1 mM *N*-acetyl-L-cysteine (Sigma-Aldrich, St. Louis, MO, USA), 2.0% Noggin/R-spondin-conditioned media, 10 μM Y-27632 (Selleck Chemicals, Houston, TX, USA), 5 µM A83-01 without EGF (HOME0) or with 0.1–50 ng/mL human recombinant EGF (Peprotech, Cranbury, NJ, USA) (HOME0.1, HOME1, or HOME50) (Table 3) unless otherwise noted. To monitor organoid growth in each well, phase-contrast images were captured by EVOS FL Cell Imaging System (Thermo Fisher scientific) and bright-field images were captured by KEYENCE Fluorescence Microscope BZ-X800 (Keyence, Osaka, Japan) equipped with the Keyence software (version 1.3.0); the latter was utilized to determine the number and size of the individual organoids (defined as ≥5000 µm^2^ spherical structures). Organoid formation rate (OFR) is calculated as the number of organoids formed per number of cells seeded and was expressed as a percentage (%) as described [32]. Cell viability was determined by trypan-blue exclusion test following trypsinization of organoids at day 11. An average number of cells constituting each organoid was determined to estimate population doubling level (PDL) by calculating log(Number of cells)−log(Number of organoids)/log2 [31,33], as each organoid represents a single-cell-derived structure formed through cell division cycles.

### 2.2. IL-13 and UAB30 Treatments

The organoids were treated from day 4 through day 11 for EN1 and day 7 through day 11 for EPC2 with or without 10 ng/mL recombinant IL-13 (200-13, Thermo Fisher Scientific, Waltham, MA, USA), reconstituted in phosphate-buffered saline (PBS) at 10 µg/mL. The organoids were treated from day 0 through day 11 with 5 µM UAB30, reconstituted in DMSO at 10 mM DMSO (V.R.A. and C.A.E., University of Alabama, Birmingham, AL, USA) or 0.05% DMSO as a vehicle (control).

### 2.3. Histology, Immunohistochemistry and Immunofluorescence

Paraffin-embedded 3D organoid products were serially sectioned for Hematoxylin and Eosin (H&E) staining, immunohistochemistry (IHC), and immunofluorescence (IF) as described previously [22]. For IHC and IF, sections were incubated with anti-Ki67 monoclonal antibody (ab16667, 1:200; Abcam, Waltham, MA, USA), anti-SOX2 monoclonal antibody (#14962, 1:300; Cell Signaling Technology (CST), Danvers, MA, USA), or anti-IVL monoclonal antibody (I9018, 1:100; Sigma-Aldrich, Burlington, MA, USA) overnight at 4 °C. For IF, Alexa Fluor^TM^ Plus 488-conjugated affinity-purified anti-mouse IgG (A32766, 1:400, Invitrogen, Waltham, MA, USA) was used for signal detection by incubating at 37 °C for 2 h, and cell nuclei were counterstained by a 4′,6-diamidino-2-phenylindole (DAPI)-containing mounting medium (H-1500-10, Vector Laboratories, Newark, CA, USA). The stained specimens were examined by Keyence automated high-resolution microscope BZ-X800 and imaged with a digital camera. The slides were evaluated by a pathologist (A.K.S.) blinded to clinical data and conditions. Whole digital-slide scans were obtained using a Leica Aperio AT2 slide scanner (Leica Biosystems, Buffalo Grove, IL, USA) at 20×. HALO AI (Indica Labs, Albuquerque, NM, USA) was trained to automatically identify the histopathological features of the organoids on the IHC slides. The organoids were automatically annotated and segmented. The levels of Ki67 and SOX2 IHC staining optical density and the number of positive cells were determined by the HALO Immunohistochemistry module per organoids by using the organoids as an object classifier. The labeling index for Ki67 and SOX2 was determined by counting at least 10 organoids per group. The chromogenic stain was quantified at the nucleus using the HALO deconvolution algorithm. The computer-generated markup image was used to confirm the specificity of the algorithm.

### 2.4. Immunoblotting

The 3D organoids isolated from Matrigel were lysed and subjected to immunoblotting as described [22,34]. Twenty μg of denatured protein was fractionated on a NuPAGE Bis-Tris 4–12% gel (Invitrogen). Following electrotransfer, the Trans-Blot Turbo Midi 0.2 µM PVDF membranes (Bio-Rad Laboratories, Hercules, CA, USA) were incubated with the primary antibodies for Phospho-EGFR (Tyr1068) (D7A5 XP^®^ Rabbit monoclonal antibody #3777, CST) at 1:1000, total EGFR (Mouse monoclonal antibody #610016; BD Transduction Laboratories, Lexington, KY, USA) at 1:1000, or anti-b-actin (#4967, 1:10,000; CST) and then with the appropriate HRP-conjugated secondary antibody (CST). β-actin served as a loading control.

### 2.5. Quantitative Reverse-Transcription Polymerase Chain Reaction (qRT-PCR)

The total RNA was isolated using GeneJET RNA Purification Kit (Thermo Fisher Scientific), and the cDNA was synthesized using High Capacity cDNA Reverse Transcription kits (Applied Biosystems, Waltham, MA, USA) according to the manufacturer’s instructions. qRT-PCR was performed with paired primers for *HBEGF*-F (5′-ATCGTGGGGCTTCTCATGT TT-3′) and *HBEGF*-R (5′-TTAGTCATGCCCAACTTCACTTT-3′), *GAPDH*-F (5′-GGAGC GAGATCCCTCCAAAAT-3′) and *GAPDH*-R (5′-GGCTGTTGTCATACTTCTCATGG-3′) using Power SYBR Green PCR Master Mix (Applied Biosystems). All the reactions were carried out on the StepOnePlus Real-Time PCR System (Applied Biosystems). The relative *HBEGF* level was normalized to *GAPDH*, serving as an internal control gene.

### 2.6. Statistical Analyses

The data were analyzed as indicated using GraphPad Prism 8.0 software. *p* < 0.05 was considered significant. The differences between two groups were analyzed by Student’s *t*-test. Comparisons of 3 or more groups were analyzed by one-way ANOVA.

## 3. Results

### 3.1. Exogenous EGF Is Dispensable for Normal Human Esophageal Organoid Growth in ADF-Based Medium

ADF-based media are commonly utilized to grow 3D organoids representing a variety of epithelial types and conditions [30,35,36,37]; however, previously reported ADF-based formulations are not particularly efficient for growing and maintaining normal human esophageal 3D organoids [22]. Given human serum EGF concentration estimated at 40 pM (i.e., ~0.25 ng/mL) [38], most ADF-based media contain supraphysiological levels (50 ng/mL) of recombinant EGF [30,35,36,37]. Excessive mitogenic stimuli may trigger cell cycle arrest or senescence in human esophageal epithelial cells [39]. We first tested how varying concentrations of recombinant EGF added to ADF-based medium, designated as human organoid medium with or without EGF (HOME) (Table 3) may influence organoid formation with extensively characterized immortalized normal human esophageal cell lines EPC1 and EPC2. Additionally, we included ADF+++ (hereafter ADF3+) [23,40] (Table 3) for a comparison, as ADF3+ was utilized to grow organoids representing human normal oral mucosa, sharing the non-cornifying stratified squamous epithelium. Likewise, we have included KSFMC [22] (Table 3) that was previously utilized to generate human normal esophageal organoids from EPC1 and EPC2 as well as endoscopic biopsies.

As visualized under bright-field microscopy and expressed as organoid formation rate (OFR), organoid formation appeared to be significantly better in HOME0 and HOME0.1 compared with HOME1 or HOME50 for both cell lines (Figure 1A–C). Likewise, EPC1 and EPC2 formed organoids poorly in ADF-based medium referred to as advanced DMEM+/+/+ (herein ADF3+) containing 50 ng/mL EGF [40] (Figure 1A–C, Table 3) that was utilized to generate 3D organoids from human oral mucosa consisting of non-cornifying stratified squamous epithelium. Moreover, OFR in HOME0 and HOME0.1 was comparable with that in KSFMC (Figure 1A–C, Table 3).

When the resulting organoids were subjected to morphological analysis by hematoxylin-eosin (H&E) staining, organoids grown in KSFMC, HOME0, or HOME0.1 displayed a normal differentiation gradient (Figure 2). By contrast, differentiation was impaired within structures grown in HOME50 or ADF3+, where degenerative changes or atypical basaloid cells were noted (Figure 2), suggesting that 50 ng/mL EGF may be excessive for normal human esophageal organoids to grow and display a proper squamous cell differentiation. Both EPC1 and EPC2 organoids grown in HOME0 or HOME0.1 had limited cornification that is more compatible with non-cornifying squamous epithelium of the esophagus than those grown in KSFMC (Figure 2).

Since endogenous EGF or other EGFR ligands may stimulate organoid cell proliferation or survival in the absence of exogenous recombinant human EGF, we tested the effect of a pharmacological EGFR tyrosine kinase inhibitor AG-1478 upon EPC1 and EPC2 organoid formation in HOME0. Interestingly, AG-1478 decreased OFR as well as cell viability in a dose-dependent manner (Figure 3A,B and Appendix A), suggesting that organoid formation requires the EGFR activity, which may be activated by EGFR ligands expressed by organoid cells in culture or present in the medium components (e.g., CM-NR; Appendix A).

### 3.2. Inhibition of TGFβ Receptor-Mediated Signaling or Rho-Associated Kinase (ROCK) Increases Organoid Formation Efficiency

In monolayer culture, TGFβ induces cell-cycle arrest to trigger senescence in EPC1 and EPC2 cells [31]. A83-01, a potent inhibitor of TGFβ receptor I, is a common component of ADF-based organoid media. However, A83-01 is not required for normal or neoplastic murine esophageal organoids to grow in the ADF-based murine organoid medium (MOM) (Table 3) [20]. We next compared HOME0 containing A83-01 and HOME0 devoid of A83-01 (HOME0^ΔA^). When EPC1 and EPC2 cells were seeded in Matrigel and fed with HOME0 and HOME0^ΔA^, OFR was found to be significantly better in the presence of A83-01 (Figure 3A,B), indicating that A83-01 is beneficial for normal human esophageal 3D organoid culture with the ADF-based medium. ROCK inhibitor Y-27632 is another common ingredient found in ADF-based organoid culture media. The Rho GTPases-ROCK pathway has functional interplays with downstream effectors SMAD2/3 to augment TGFβ receptor-mediated signaling [41,42]. When ROCK was inhibited by Y-27632 in HOME0, both EPC1 and EPC2 cells exhibited an improved organoid formation rate (Figure 3A,B), confirming the beneficial effect of Y-27632 compound in human esophageal organoid culture utilizing the ADF-based medium.

### 3.3. HOME0 Permits Normal Esophageal Organoid Formation from Patient Biopsies

Given improved OFR and the differentiation gradient formation for EPC1 and EPC2 cells in HOME0 (Figure 1 and Figure 2), we further tested HOME0 on PDO formation with normal esophageal endoscopic biopsies. Following dissociation, biopsy-derived cells were processed for organoid culture, either immediately (EN1) or following cryopreservation (EN2 and EN3). Testing HOME0 and three other media, HOME50, ADF3+, and KSFMC (Table 3), all the biopsy samples (EN1–3) gave rise to spherical structures. When these normal PDOs were subjected to morphological analyses, an inward differentiation gradient was observed in structures formed in KSFMC or HOME0 (Figure 4). Cornification, normally absent in normal human esophageal squamous epithelium, was present in the organoids grown in KSFMC, but limited, if any, in HOME0. Organoids grown in HOME50 or ADF3+ displayed either nuclear atypia or degeneration, suggesting that cells may be under a certain stress in these medium conditions.

While none of these media permitted indefinite growth, replicative lifespan or the total number of cell division measured as population doubling level (PDL) appeared to be higher in KSFMC and HOME0 than HOME50 and ADF3+ (Appendix A–C). The maximum PDL of these normal mucosa-derived organoids was 40–60 when grown in KSFMC and HOME0 (Appendix A), compatible with the PDL for normal human esophageal epithelial primary monolayer culture where cells cease proliferation and undergo senescence at the end of their replicative lifespan aka the Hayflick limit [31,33]. The PDL was less than 20 when grown in HOME50 or ADF3+ (Appendix A). Interestingly, OFR was found to be the highest at passage 1 for all the clinical samples, especially in HOME0 and KSFMC, although OFR decreased as passaged in subculture (Appendix A). While initial organoid culture may contain biopsy-derived non-epithelial cells that may account for a lower OFR, the transient elevation of OFR at the first passage (P1) suggests that esophageal organoids contain a subset of epithelial cells that underwent renewal to initiate organoids in subsequent passages.

### 3.4. IL-13 Induces BCH-like Changes in Normal Esophageal Organoids Formed in HOME0

We next explored the utility of HOME0-grown normal human esophageal 3D organoids as an experimental platform to model pathologic epithelial conditions. We have previously modeled EoE-related reactive epithelial changes and BCH where human esophageal 3D organoids were grown in KSFMC and stimulated with EoE-relevant cytokines including interleukin (IL)-13 [22,32]. In normal esophageal PDOs, EN1 and EPC2 grown in HOME0 stimulation with recombinant IL-13 resulted in the formation of less differentiated structures compatible with BCH as evidenced by an expansion of basaloid cells expressing basal cell-specific transcription factor SOX2 by immunohistochemistry (IHC) and IVL, an early-stage squamous-cell differentiation marker by immunofluorescence (IF) (Figure 5A). BCH was corroborated by significant upregulation of SOX2 and downregulation of IVL within the basaloid cell compartment at the outermost cell layer of EN1 organoids, as documented by AI-assisted scoring (Appendix A and Figure 5B–D). However, these molecular changes were quantitatively not as robust in EPC2 organoids (Figure 5E–G), suggesting differential cytokine sensitivity between EN1 and EPC2. Moreover, Ki67, a proliferation marker, was not significantly elevated in either EN1 or EPC2 with BCH-like changes (Figure 5B,E), suggesting that IL-13 delayed post-mitotic terminal differentiation but did not necessarily stimulate basal cell proliferation in a persistent manner.

### 3.5. RXR Stimulation Induces BCH-like Changes in Normal Esophageal Organoids Formed in HOME0

EoE patient esophageal biopsies show upregulation of retinoic acid receptor (RAR)-β mRNA expression [43,44]. In keratinocytes, biological effects of vitamin A are mediated by heterodimers of RARs and retinoic-X receptors (RXRs) that act as a transcription factor [45]. The effect of activation of retinoic acid signaling in the human esophageal epithelium remains elusive. We stimulated EPC2 organoids and a normal human PDO line EN1 with UAB30, a potent synthetic RXR agonist that enhances all-trans retinoic acid signaling in human epidermal keratinocytes in organotypic 3D culture [46]. In normal esophageal PDO EN1 and EPC2 organoids grown in HOME0, UAB30 induced BCH-like epithelial changes with increased Ki67-positive basaloid cells while decreasing IVL expression concurrently (Figure 6A–G). Since retinoic acid-induced epidermal hyperplasia involves heparin-binding EGF-like growth factor (HB-EGF), an EGFR ligand [47,48,49,50], we next evaluated HB-EGF expression and the EGFR activity. UAB30 upregulated HB-EGF mRNA expression (Appendix A) as well as the EGFR tyrosine phosphorylation to an extent that was comparable with the effect of 1 ng/mL EGF (Appendix A), suggesting that HB-EGF may activate EGFR to mediate UAB30-stimulated BCH.

In aggregate, the above findings indicate that the normal proliferation–differentiation gradient and its perturbation are recapitulated in human esophageal organoids grown in the HOME0 medium conditions, thus establishing ADF-based organoid culture medium conditions.

## 4. Discussion

In optimizing medium conditions to generate normal human esophageal organoids utilizing ADF medium, we found that the removal of exogenous EGF and the addition of A83-01, a TGFβ receptor I inhibitor, is most helpful to improve organoid formation efficiency. Our optimized HOME0 medium contains a limited number of supplements compared to other ADF-based media such as ADF3+ (Table 3), described previously [11,40]. The HOME0-grown organoids displayed a well-organized proliferation–differentiation gradient that recapitulates the histologic architecture of the stratified squamous epithelium of the normal human esophagus. Furthermore, HOME0 medium appeared to be permissive for modeling BCH that represents reactive epithelial changes common in EoE, gastroesophageal reflux diseases, and other esophageal disease conditions.

One remarkable premise is that exogenous EGF is dispensable for HOME0 and that addition of EGF at supraphysiological concentration (50 ng/mL) is detrimental for optimal organoid growth of human normal esophageal epithelial cells. This may explain why previously reported ADF-based media including ADF3+ performed poorly compared to other medium formulations such as KSFMC containing 1 ng/mL recombinant human EGF that is closer to a human serum level [38]. Our data do not preclude the importance of EGFR signaling for normal human esophageal organoid formation in HOME0, as EGFR tyrosine phosphorylation was detectable in the absence of exogenous EGF (Appendix A) and EGFR tyrosine kinase inhibitor AG-1478 decreased organoid formation and cell viability (Figure 3 and Appendix A), reinforcing the role of endogenous EGFR activity in organoid formation. However, 0.1 ng/mL EGF added into HOME0 did not induce EGFR phosphorylation significantly compared to HOME0 (control) (Appendix A), suggesting that a minimal and a physiologically-relevant level of EGFR ligands (e.g., ~0.25 ng/mL EGF in the serum [38]) is sufficient for human normal esophageal organoid formation in HOME0.

Interestingly, murine normal esophageal organoids grow well in either KSFMC or ADF-based media, including our own mouse organoid medium (MOM) containing 50 ng/mL EGF (Table 3) [11,20], suggesting species differences in EGFR signaling and EGF sensitivity in esophageal epithelial cells. Given the oncogenic role of EGFR in esophageal carcinogenesis, normal human esophageal cells may negate malignant transformation via degenerative changes in response to excessive mitogenic stimuli as observed in HOME50 (Figure 1). In agreement, EGFR overexpression alone induces TGFβ-mediated senescence in EPC1 and EPC2 cells while concurrent inactivation of cell-cycle checkpoint functions is permissive for transformation with epithelial-mesenchymal transition [39]. Since we did not compare ADF3+ with or without EGF in this study, we cannot exclude the possibility that the benefit we observe by HOME0 may be accounted for by the lack of other factors and agents present in ADF3+ (e.g., FGFs and CHIR99021). However, they are not required for murine organoid culture in MOM [20] (Table 3). Additionally, the lack of ADF3+ components in HOME50 does not account for the degenerative morphological changes (Figure 2 and Figure 4), reduced OFR and shorter PDL (Figure 1 and Appendix A) which were observed in both ADF3+ and HOME50 compared to HOME0 and KSFMC. Unlike ADF3+, HOME media contain the N2 supplement [51] consisting of insulin, transferrin, progesterone, selenium, and putrescine, which are all redundant factors present in B27 [52] and the ADF-base medium. We are currently testing HOME0 and HOME50 devoid of the N2 supplement without substantial differences (unpublished observations). Therefore, we may establish a simplified ADF3+ to identify a minimal standard media formulation except for the increased A83-01 concentration.

As A83-01 significantly improved organoid formation, this study also highlights that TGFβ may act as a potential safeguard mechanism against aberrant proliferation of normal human esophageal epithelial cells exposed to excessive mitogenic stimuli. A83-01 is not required in KSFMC medium to form normal human esophageal organoids [22]. Unlike KSFMC, however, ADF and its common supplements such as N2 and B27 contain certain agents and factors redundantly. Amongst them, insulin (>10–15 µg/mL altogether) stimulates IGF-I receptor at such supraphysiologic concentrations [3,53] as to provide a mitogenic stimulus as well as a cytoprotection [54], the latter via the PI3K-AKT pathway. In the absence of exogenous EGF, IGF-I stimulates esophageal basal cell proliferation [53]. It is plausible that A83-01 limits TGFβ-mediated cell cycle arrest in normal human esophageal keratinocytes under a variety of stressors [31,39]. In line with previous attempts [22], all biopsy-derived normal human esophageal PDOs ceased proliferation within 40–60 PDLs regardless of the medium (Figure 5), suggesting that pharmacological inhibition of TGFβ receptor I-mediated signaling does not prevent telomere attrition and replicative senescence [33]. Y-27632, a selective inhibitor of Rho-associated, coiled-coil containing protein kinase and a common ingredient of ADF-based organoid media, may immortalize keratinocytes [54]. Y-27632 improved growth and OFR of hTERT-immortalized EPC1 and EPC2 cells (Figure 3) but did not extend the replicative lifespan of non-immortalized primary esophageal keratinocytes grown in PDOs EN1-EN3 (Figure 5). In fact, organoids made with immortalized EPC1 and EPC2 cells grew indefinitely in HOME0. Interestingly, murine normal esophageal organoids grow indefinitely (i.e., over 20 passages, exceeding 250 PDLs) in ADF-based medium without A83-01 [20], highlighting another species difference.

Our optimized organoid culture condition in HOME0 recaptured a proper proliferation–differentiation gradient in the normal stratified squamous epithelium of the esophagus as well as perturbation relevant to esophageal epithelial pathologies. Normal human esophageal organoids grown in HOME0 display a modest, if any, cornification compared to those grown in KSFMC (Figure 2 and Figure 5). HOME0 may excel KSFMC in this regard, as the human oral/esophageal squamous epithelium does not normally undergo cornification, unlike the epidermis or the oral/esophageal squamous epithelia in mice. Keratinocyte differentiation depends on calcium [55,56]. Although calcium is a key ingredient that permits esophageal organoid formation and squamous cell differentiation in KSFMC [22], calcium does not account for limited cornification in HOME0 because ADF contains a higher calcium concentration than KSFMC (Table 3).

As terminal differentiation occurs in post-mitotic cells, the differential quantity of mitogenic factors and hormones such as transferrin, insulin, and vitamin A present in these media may account for the differences in degree of observed differentiation. Although the content of KSFM is not fully disclosed by the manufacturer, it is possible that retinoic acid signaling by vitamin A in the B27 supplement added to HOME0 may delay terminal differentiation, as was enhanced by UAB30 (Figure 6). Both UAB30 and IL-13 had similar and profound impacts upon the differentiation gradient through basaloid cell expansion and delayed terminal differentiation (Figure 5 and Figure 6). Interestingly, UAB30 did not induce degenerative changes observed in the presence of 1 ng/mL EGF (Figure 2). Nevertheless, the extent of the UAB30-induced EGFR phosphorylation was comparable to the effect of 1 ng/mL EGF (Appendix A), suggesting that UAB30 and EGF have differential biological effects. EGFR limits terminal differentiation by suppressing Notch signaling [14]. Our data indicate that intrinsic EGFR ligands such as HB-EGF may contribute to BCH. Interestingly, IL-13 and other EoE-related cytokines suppress Notch activation [22]. In macrophages, Notch signaling promotes differentiation that is antagonized by retinoic acid signaling [57]. Further studies are warranted to interrogate functional interplays between these cytokines and retinoic acid signaling in EoE-related BCH.

Limitations of this study include the largely empirical nature of our study design, where EGF dosage and TGFβ receptor I inhibition were tested without genetic or pharmacological screening to identify candidate factors and signaling pathways that may be utilized to improve organoid culture conditions. It remains unknown whether HOME0 supports organoid formation from non-esophageal squamous epithelial cells (e.g., head-and-neck/oral epithelium) as well as neoplastic human esophageal cells. Optimization and characterization of preneoplastic and neoplastic human organoids is underway in our laboratory. Additionally, it remains elusive as to what factors and mechanisms may govern human–mouse species differences with respect to EGF sensitivity, A83-01 requirement, replicative lifespan, and degree of terminal differentiation (i.e., cornification) in normal esophageal organoids. We have noted that the baseline OFR vary by 2–3-fold differences due to cell culture conditions and medium components (e.g., seeding cell density variability and the Matrigel lot-to-lot variability). When OFR deviates substantially, a mycoplasma test is warranted. Finally, we could generate patient-derived esophageal organoids utilizing HOME0 from endoscopic biopsies from an adult (CUIMC) and children (CHOP), either fresh or cryopreserved prior to tissue dissociation. However, the total number of the samples in this study is small. The expansion of the biorepository of human esophageal organoids representing normal and non-neoplastic esophageal conditions (e.g., EoE and GERD) is underway at the OCCC of the Columbia University Digestive and Liver Disease Research Center at the CUIMC.

## 5. Conclusions

In summary, we have established a HOME0 medium formulation and validated its utility to grow single-cell-derived normal human esophageal organoids, enabling the comparison of species differences in esophageal epithelial characteristics as well as organoids grown in the broadly utilized ADF-based medium supplemented with different factors and agents.

## Figures and Tables

**Figure 1 biomolecules-14-01126-f001:**
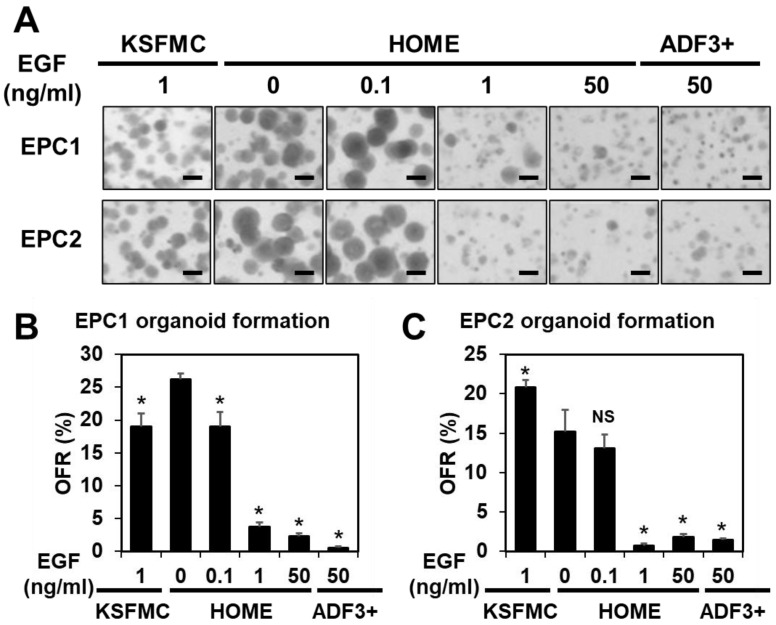
Low EGF concentrations (≤0.1 ng/mL) may facilitate organoid formation in ADF-base medium. EPC1 and EPC2 cells were seeded at a density of 5000 cells per 50 µL Matrigel and fed with KSFMC, HOME, or ADF3+, containing human recombinant EGF at indicated final concentrations and allowed to form organoids. (**A**) Resulting structures were captured by Keyence and photomicrographed. Scale bar, 200 µm. (**B**,**C**) OFR was determined and plotted in graphs. *, *p* < 0.05 vs. 0 ng/mL EGF in HOME (i.e., HOME0); NS, not significant vs. HOME0, n = 3 per group. One-way ANOVA was used for the overall comparison.

**Figure 2 biomolecules-14-01126-f002:**
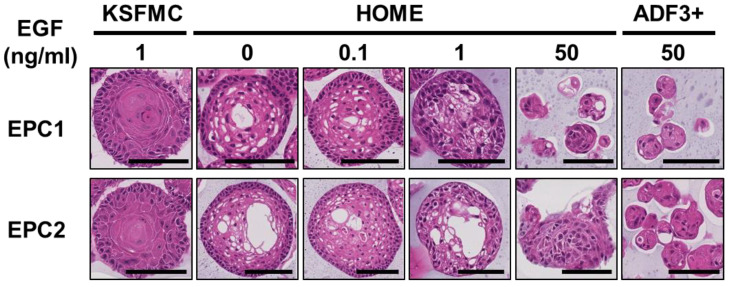
EPC1 and EPC2 organoids were grown in KSFMC, HOME, or ADF3+, containing recombinant human EGF at indicated final concentrations and subjected to H&E staining. Structures grown in KSFMC, HOME0, or HOME0.1 displayed concentric layers of spinous-like cells with a squamous cell differentiation gradient with a keratinized material to a partial (HOME0 or HOME0.1) or a total extent (KSFMC). In HOME50 and ADF3+, structures tended to display small clusters of cells (<32 cells in a cross-sectional plane) often with debris, largely involutional or atrophic. Atypical basaloid cells (mild to moderate) were noted in structures grown in HOME1, HOME50, and ADF3+. Scale bar, 100 µm.

**Figure 3 biomolecules-14-01126-f003:**
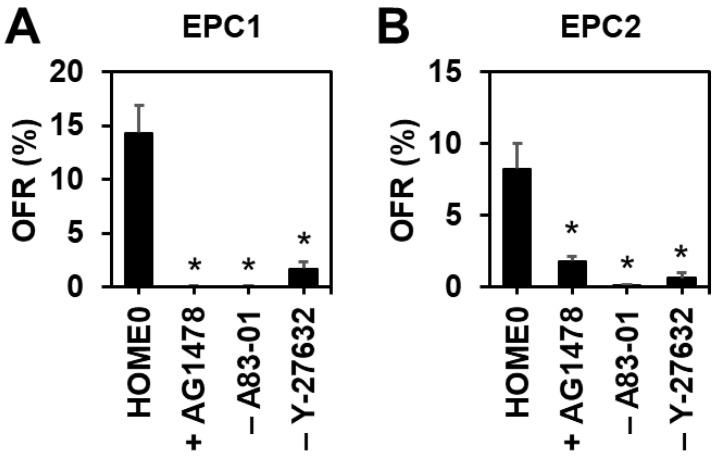
EPC1 and EPC2 organoid formation is suppressed in the presence of AG-1478, an EGFR inhibitor (100 nM), or in the absence of either A83-01, an inhibitor for TGFβ receptor-mediated signaling or Y-27632, an inhibitor for ROCK-mediated signaling. Note that HOME0 contains 5 µM A83-01 and 10 µM Y-27632. (**A**), EPC1; (**B**), EPC2. *, *p* < 0.05 vs. HOME0, n = 3 in (**A**,**B**). One-way ANOVA was used for the overall comparison.

**Figure 4 biomolecules-14-01126-f004:**
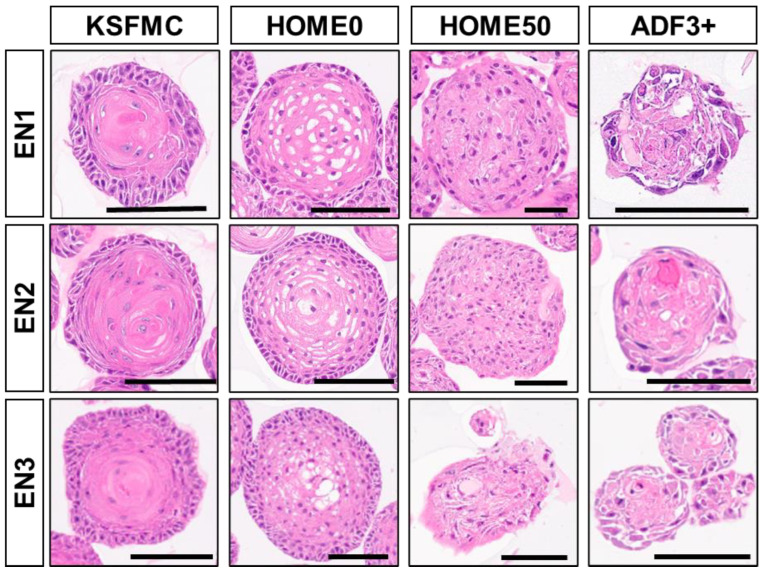
Normal esophageal PDOs recapitulate squamous epithelial differentiation gradient when grown in KSFMC or HOME0. Normal esophageal PDOs (EN1–3) grown with indicated media were subjected to H&E staining at the end of passage 1. Note that structures grown in KSFMC and HOME0 had concentric layers of spinous-like cells with a differentiation gradient with a keratinized material to a partial (HOME0) or a total extent (KSFMC). Structures grown in HOME50 or ADF3+ displayed no apparent differentiation gradient with involutional changes or mild atypia. Scale bars, 100 µm.

**Figure 5 biomolecules-14-01126-f005:**
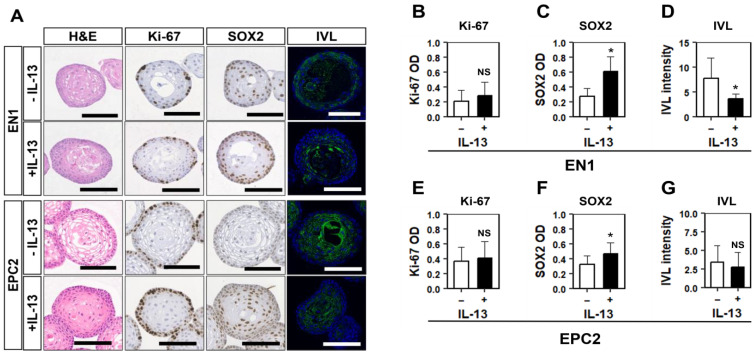
IL-13 induces BCH-like epithelial changes in normal human esophageal organoids grown in HOME0. Established organoids were stimulated with or without recombinant human IL-13 and subjected to H&E staining, IHC for indicated markers, or IF for IVL (**A**). Halo AI and multiplex IHC modules were used for organoid segmentation, and Halo IHC was used for quantification of optical density (OD) (**B**,**C**,**E**,**F**). The OD average was calculated in each organoid. For IF, nuclei were counterstained with DAPI. IVL intensity was measured by ImageJ (https://imagej.net/) for each organoid (**D**,**G**). * Indicates *p* < 0.05, NS: non-significant. Student’s *t*-test was used for the comparison. Scale bars, 100 µm.

**Figure 6 biomolecules-14-01126-f006:**
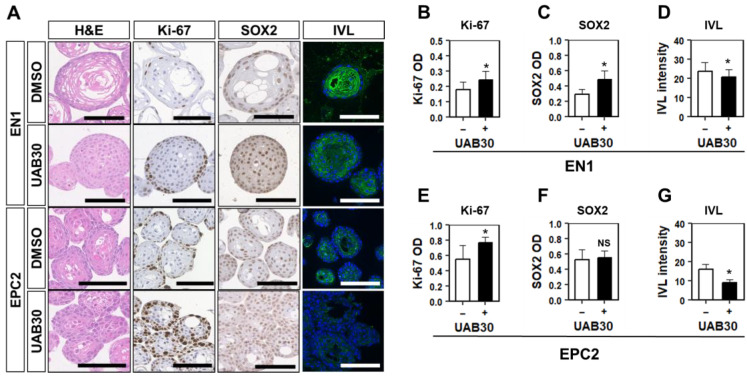
UAB30 induces BCH-like epithelial changes in normal human esophageal PDO and EPC2 organoids grown in HOME0. Organoids were grown in the presence or absence of 5 µM UAB30 and to H&E staining, IHC for indicated markers, or IF for IVL (**A**). Halo AI and multiplex IHC modules were used for organoid segmentation, and Halo IHC was used for quantification of optical density (OD) (**B**,**C**,**E**,**F**). The OD average was calculated in each organoid. For IF, nuclei were counterstained with DAPI. IVL intensity was measured by ImageJ (https://imagej.net/) for each organoid (**D**,**G**). * Indicates *p* < 0.05, NS: non-significant. Student’s *t*-test was used for the comparison. Scale bars, 100 µm.

**Table 1 biomolecules-14-01126-t001:** Abbreviations of general terms.

Abbreviation	Name
AI	Artificial Intelligence
BCH	basal cell hyperplasia
BMP	bone morphogenetic proteins
DAPI	4′,6-diamidino-2-phenylindole
DMSO	Dimethyl sulfoxide
EoE	eosinophilic esophagitis
H&E	Hematoxylin and Eosin
IF	immunofluorescence
IHC	immunohistochemistry
IL	interleukin
IVL	Involucrin
OFR	Organoid formation rate
PBS	phosphate-buffered saline
PDL	population doubling level
PDO	Patient-derived organoids
qRT-PCR	quantitative reverse-transcription polymerase chain reaction
RAR	retinoic acid receptor
RXR	retinoic-X receptor
3D	three-dimensional
TGF-β	transforming growth factor-beta

**Table 2 biomolecules-14-01126-t002:** Abbreviations of cell culture media and related growth factors.

Abbreviation	Name
ADF	advanced Dulbecco’s Modified Eagle Medium/Ham’s F-12
ADF3+	advanced DMEM +/+/+
CM	conditioned medium
CM-NR	CM containing Noggin and R-Spondin1
EGF	epidermal growth factor
FGF2	fibroblast growth factor 2
FGF10	fibroblast growth factor 10
HB-EGF	heparin-binding EGF-like growth factor
HOME	human organoid medium containing EGF at indicated concentrations
HOME0	human organoid medium containing 0 ng/mL EGF
HOME0.1	human organoid medium containing 0.1 ng/mL EGF
HOME1	human organoid medium containing 1 ng/mL EGF
HOME10	human organoid medium containing 10 ng/mL EGF
HOME0 ^ΔA^	HOME0 devoid of A83-01
KSFM	keratinocyte serum-free medium
KSFMC	KSFM medium containing 0.6 mM Ca^2+^
MOM	murine organoid medium

**Table 3 biomolecules-14-01126-t003:** Organoid media comparison.

Name	ADF3+	HOME	HOME	MOM	KSFMC
Base medium	ADF	ADF	ADF	ADF	KSFM
B27	(+)	(+)	(+)	(+)	U
N2	(−)	(+)	(+)	(+)	U
NAC	(+)	(+)	(+)	(+)	U
CM-NR	(+)	(+)	(+)	(+)	U
EGF (ng/mL)	50 ^†^	0 or 50 ^†^	Various ^†^ (0–50)	50 ^††^	1 ^†^
A83-01 (µM)	0.5	0	5	0	U
Nicotinamide	(+)	(−)	(−)	(−)	U
FGF110	(+)	(−)	(−)	(−)	U
FGF2	(+)	(−)	(−)	(−)	U
CHI99021	(+)	(−)	(−)	(−)	U
Forskolin	(+)	(−)	(−)	(−)	U
Prostaglandin E2	(+)	(−)	(−)	(−)	U
CaCl_2_ (mM)	1	1	1	1	0.6
Y-27632	(+)	(+)	(+)	(+)	(+)

ADF contains GlutaMAX, HEPES and antibiotics. A83-01, inhibitor for TGF-β receptor-mediated signaling. CHIR-99021, Glycogen synthase kinase-3 inhibitor. None of the ADF-based media in Table 3 contains WNT3A, Gastrin, and SB202190, the agents often used to grow organoids of non-squamous epithelial origin. Y-27632, is added for 3 days only at the onset of organoid culture/subculture or may be added continuously. U, undisclosed. †, recombinant human EGF; ††, recombinant mouse EGF.

## Data Availability

The original contributions presented in the study are included in the article/Appendix A; further inquiries can be directed to the corresponding author.

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
