# Peer review of "Modeling Epithelial Homeostasis and Perturbation in Three-Dimensional Human Esophageal Organoids"

_biomolecules, 2024, doi:10.3390/biom14091126_

Round 1
Reviewer 1 Report
Comments and Suggestions for Authors
The authors optimized culture media for human normal esophageal organoids and found that exogenous EGF concentration was key factor to make balance between proliferation and differentiation of esophageal epithelium. The authors also found that inhibition of TGFbeta and ROCK signaling is additional factor for maintenance of organoids and that a RXR agonist induces basal cell hyperplasia via enhanced endogenous EGF signals. Some issues should be clarified to improve the manuscript.
1. Abbreviations used in the manuscript are hard to find, especially names for culture media, i.e. HOME0. Please list abbreviations or add full spells of culture media in Table 1 figure legend.
2. OFR of EPC1 under HOME0 was 10-25% in Fig. 2B and Fig. 3C, E, but somehow very low in Fig. 3A (only 1.5%). Is any reason to explain this difference, such as vehicle effect, experimental conditions, etc.?
3. Comparisons for 3 or more groups should be analyzed by multiple comparison such as one-way ANOVA, not by t-test, in Fig. 2B, C and Fig. 3A, B.
4. The results suggest that reduction of EGF in culture media is a key factor to maintain ‘normal’ differentiation of esophageal organoids and that higher concentration of EGF increases proliferation and reduces differentiation. In contrast, the RXR agonist UAB30 induced HB-EGF-EGF receptor signals, causing basal cell hyperplasia. How much concentration of EGF is consistent with the effects of UAB30? The authors may add Ki67, SOX2 and IVL staining and analysis in HOME0-50 samples in Fig. 2.
Author Response
The following is a point-by-point response to the helpful and insightful comments by Reviewer 1.
“The authors optimized culture media for human normal esophageal organoids and found that exogenous EGF concentration was key factor to make balance between proliferation and differentiation of esophageal epithelium. The authors also found that inhibition of TGFbeta and ROCK signaling is additional factor for maintenance of organoids and that a RXR agonist induces basal cell hyperplasia via enhanced endogenous EGF signals. Some issues should be clarified to improve the manuscript”.
We agree and have addressed these concerns as below.
- “Abbreviations used in the manuscript are hard to find, especially names for culture media, i.e. HOME0. Please list abbreviations or add full spells of culture media in Table 1 figure legend.”
We have now provided the lists of abbreviations as Tables 1 and 2 for general terms and cell culture media/growth factors, respectively, as indicated in line 78. We have revised Table 3 (original Table 1) and its footnote.
- “OFR of EPC1 under HOME0 was 10-25% in Fig. 2B and Fig. 3C, E, but somehow very low in Fig. 3A (only 1.5%). Is any reason to explain this difference, such as vehicle effect, experimental conditions, etc.?”
While the baseline OFR vary 2-3-fold differences due to cell culture conditions and medium components (e.g., seeding cell density and the Matrigel lot-to-lot variability), we have realized mycoplasma contamination in both EPC1 and EPC2 cells utilized in the experiments shown in original Fig. 3. We apologize for this oversight. After thawing frozen stocks made at earlier passages, we have validated the mycoplasma negative status for both EPC1 and EPC2 cells. They showed the baseline OFRs (8-15%) that were more compatible with the data shown in original Fig. 1. In new experiments, both cell lines showed significantly decreased OFR in the presence of AG1478 (EGFR inhibitor) and in the absence of either A83-01 (TGFβ receptor-mediated signaling) or Y-27632 (ROCK inhibitor), reproducing the data in original Fig. 3. We have now revised Fig. 3 with new data with a higher baseline OFR for both EPC1 and EPC2. In the Materials and Methods section 2.1, we have added description about mycoplasma testing and cell line authentication (lines 134-136). In the Discussion section, we have discussed the potential factors including mycoplasma that may influence the OFR (lines 514-517).
- “Comparisons for 3 or more groups should be analyzed by multiple comparison such as one-way ANOVA, not by t-test, in Fig. 2B, C and Fig. 3A, B”.
We have now performed one-way ANOVA for comparisons of 3 or more groups and revised the methods section 2.6 (line 243) and indicated in figure legends.
- “The results suggest that reduction of EGF in culture media is a key factor to maintain ‘normal’ differentiation of esophageal organoids and that higher concentration of EGF increases proliferation and reduces differentiation. In contrast, the RXR agonist UAB30 induced HB-EGF-EGF receptor signals, causing basal cell hyperplasia. How much concentration of EGF is consistent with the effects of UAB30? The authors may add Ki67, SOX2 and IVL staining and analysis in HOME0-50 samples in Fig. 2”
We could not perform immunostaining of organoids grown in the presence of 1 ng/mL EGF due to a low yield of mature structures as indicated by low OFR (Fig. 1B, C). We have now confirmed that UAB30 (5 µM) induced EGFR phosphorylation to an extent that was comparable with the effect of 1 ng/mL EGF (Supplemental Fig. S4B and C). Nevertheless, UAB30 did not induce degenerative changes observed in the presence of 1 ng/mL EGF (Fig. 2), suggesting that UAB30 and EGF have differential biological effects. These observations were described in the results section 3.5 (lines 393-394) and the extended discussion (lines 425-429; lines 493-497).

Reviewer 2 Report
Comments and Suggestions for Authors
Major comments:
The organoid formation rate (OFR) considers the number of organoids. However, figure 1A appears to show HOME0 produces larger organoids but not clearly more organoids. Figure 1C suggests that roughly 10 times more organoids should be visible in HOME0 compared to ADF3+50. However, it seems the same amount of organoids is present, but these are larger when grown in HOME0. Should you not also take organoid size into account besides formation rate? Could you provide more representative (larger) examples?
OFR of EPC1 in HOME0 in figure 1B is 25%, in figure 3 it ranges from 0 to 4 %. For EPC2 in 1C OFR is 15%, in figure 3 it ranges from 1-9%. What explains these huge differences in formation rate at baseline, given these are cell lines? The error bar in figure 1B is very small, just as in figure 3C. However, their point estimates of the OFR are 25% and 4%.
Minor comments:
The statement you aim to prove in Line 221 is whether the concentration of recombinant EGF influences organoid formation in both cell lines. Could you explain why the KSFMC and ADF3+ groups were included in the comparison?
For Figures 1B and 1C, is it appropriate to compare HOME0 with KSFMC1 and ADF3+ 50? What conclusions can be drawn from this experiment given the numerous differences (and some unknown differences in the case of FSFMC) between these media conditions?
In Line 270, if you suggest that organoid formation requires EGFR activity which may be activated by EGFR ligands present in the medium component (NR-CM). Could you add a figure in the supplementary materials showing the results with and without these components to substantiate this claim?
EN1 differs from EN2 and 3 in multiple respects being exposition to crypreservation and derivation from an adult versus from children. The last aspect is not reflected in section 3.3 of the results.
Comments on the Quality of English Language100 not only the normal squamous
Caption figure 6: p<0.0-5, remove "-"
Author Response
The following is a point-by-point response to the helpful and insightful comments of the Reviewer 2:
Reviewer 2
- “The organoid formation rate (OFR) considers the number of organoids. However, figure 1A appears to show HOME0 produces larger organoids but not clearly more organoids. Figure 1C suggests that roughly 10 times more organoids should be visible in HOME0 compared to ADF3+50. However, it seems the same amount of organoids is present, but these are larger when grown in HOME0. Should you not also take organoid size into account besides formation rate? Could you provide more representative (larger) examples?”
We agree and now included representative bright-field organoids images taken at a higher magnification (revised Fig. 1A). The size of structures grown in HOME1, HOME50 and ADF3+ was smaller than those grown in KSFMC, HOME0 or HOME1. Most of these small structures did not meet the 5,000 µm2 threshold to be counted as mature organoids, reflecting upon a significantly lower OFR for HOME1, HOME50 and ADF3+ (Fig. 1B and C). Hematoxylin-Eosin (H&E) staining (Fig. 2) confirmed that most small structures were degenerative.
- “OFR of EPC1 in HOME0 in figure 1B is 25%, in figure 3 it ranges from 0 to 4 %. For EPC2 in 1C OFR is 15%, in figure 3 it ranges from 1-9%. What explains these huge differences in formation rate at baseline, given these are cell lines? The error bar in figure 1B is very small, just as in figure 3C. However, their point estimates of the OFR are 25% and 4%”.
While the baseline OFR vary 2-3-fold differences due to cell culture conditions and medium components (e.g., seeding cell density and the Matrigel lot-to-lot variability), we have realized mycoplasma contamination in both EPC1 and EPC2 cells utilized in the experiments shown in original Fig. 3. We apologize for this oversight. After thawing frozen stocks made at earlier passages, we have validated the mycoplasma negative status for both EPC1 and EPC2 cells. They showed the baseline OFRs (8-15%) that were more compatible with the data shown in original Fig. 1. In new experiments, both cell lines showed significantly decreased OFR in the presence of AG1478 (EGFR inhibitor) and in the absence of either A83-01 (TGFβ receptor-mediated signaling) or Y-27632 (ROCK inhibitor), reproducing the data in original Fig. 3. We have now revised Fig. 3 with new data with a higher baseline OFR for both EPC1 and EPC2. In the Materials and Methods section 2.1, we have added description about mycoplasma testing and cell line authentication (lines 134-136). In the Discussion section, we have discussed the potential factors including mycoplasma that may influence the OFR (lines 514-517).
Minor Comments
- “The statement you aim to prove in Line 221 (revised line 255-) is whether the concentration of recombinant EGF influences organoid formation in both cell lines. Could you explain why the KSFMC and ADF3+ groups were included in the comparison?”
We apologize for the lack of clarity about why KSFMC and ADF3+ were included here. We included them as positive controls as utilized previously to grow normal human esophageal organoids (Kasagi et al. Cell Mol Gastroenterol Hepatol. 2018, PMID: 29552622) and normal human oral organoids (Driehuis et al. Cancer Discov. 2019, PMID: 31053628; Driehuis et al. Nat Protoc. 2020, PMID: 32929210), respectively. Human esophageal organoids grew well in both KSFMC and HOME0 although the resulting structures were more keratinized in KSFMC than HOME0 (Fig. 2) as described in the Results section 3.1 (lines 278-280) and discussed in lines 477-481 in the revised manuscript. Since the human esophageal epithelium shows no keratinization in the superficial cell layer, we concluded that HOME0 may better recapitulate normal human esophageal epithelium than KSFMC. We included ADF3+, in part because both oral and esophageal mucosa share the similar non-cornifying stratified squamous epithelia and in part because ADF3+ and HOME50 share the same base medium containing 50 ng/mL EGF although ADF3+ contains more various growth factors and agents than HOME50 (revised Table 3). In both ADF3+ and HOME50, esophageal organoids showed degenerative changes, suggesting that the lack of ADF3+ components did not cause degenerative changes observed in HOME50. We have added these points in the revised Results section 3.1 (lines 259-264) and revised Discussion section (lines 439-449).
- “For Figures 1B and 1C, is it appropriate to compare HOME0 with KSFMC1 and ADF3+ 50? What conclusions can be drawn from this experiment given the numerous differences (and some unknown differences in the case of FSFMC) between these media conditions?”
Fig. 1B and C indicated that EPC1 and EPC2 formed organoids at comparable rates in HOME0 and KSFMC1, but not ADF3+, suggesting that differences in medium components (e.g., KSFMC vs. HOME0) do not necessarily affect organoid formation while influencing morphological characteristics to some extent (Fig. 2) as discussed above for the Minor comment #1.
- “In Line 270 (revised line 288), if you suggest that organoid formation requires EGFR activity which may be activated by EGFR ligands present in the medium component (NR-CM). Could you add a figure in the supplementary materials showing the results with and without these components to substantiate this claim?”
We have now included new data showing that CM containing Noggin and R-Spondin1 increased the OFR (Supplemental Fig. S1B) corroborating this premise and described in the revised Results section (line 288).
- “EN1 differs from EN2 and 3 in multiple respects being exposition to cryopreservation and derivation from an adult versus from children. The last aspect is not reflected in section 3.3 of the results.”
Although we could grow patient-derived normal esophageal organoids from these samples (Figure 4) as described in the result section 3.3, the sample size was too small to draw statistically meaning conclusions about the impact of the conditions of biopsy specimen (i.e., fresh vs. cryopreserved) and patients (i.e., children vs. adults). We described how biopsies were processed in section 2.1 (lines 142-147) and discussed as limitations of this study in the Discussion section (lines 517-523).
Comments on the Quality of English Language:
‘not only the normal squamous’ and ‘Caption figure 6: p<0.0-5, remove "-"’
Thank you for catching these errors. We have corrected both in the revised manuscript.
We believe that the manuscript is much improved and are appreciative of the helpful comments of the Reviewers. We hope you find the revised manuscript suitable for publication.
